# Automatic catheter detection in pediatric X-ray images using a scale-recurrent network and synthetic data

**Xin Yi**
Department of Medical Imaging
University of Saskatchewan
SK, CA, S7N 5E5
xin.yi@usask.ca

**Scott Adams**
Department of Medical Imaging
University of Saskatchewan
SK, CA, S7N 0W8
scott.adams@usask.ca

**Paul Babyn**
Department of Medical Imaging
University of Saskatchewan
SK, CA, S7N 0W8
Paul.Babyn@saskhealthauthority.ca

**Abdul Elnajmi**
Department of Medical Imaging
University of Saskatchewan
SK, CA, S7N 0W8
abe823@mail.usask.ca

## Abstract

Catheters are commonly inserted life supporting devices. X-ray images are used to assess the position of a catheter immediately after placement as serious complications can arise from malpositioned catheters. Previous computer vision approaches to detect catheters on X-ray images either relied on low-level cues that are not sufficiently robust or only capable of processing a limited number or type of catheters. With the resurgence of deep learning, supervised training approaches are begining to showing promising results. However, dense annotation maps are required, and the work of a human annotator is hard to scale. In this work, we proposed a simple way of synthesizing catheters on X-ray images and a scale recurrent network for catheter detection. By training on adult chest X-rays, the proposed network exhibits promising detection results on pediatric chest/abdomen X-rays in terms of both precision and recall.

## 1 Introduction

Catheters and tubes, including endotracheal tubes (ETTs), umbilical arterial catheters (UACs), umbilical venous catheters (UVCs), and nasogastric tubes (NGTs), are commonly used in the management of critically ill or very low birth weight neonates [8]. For example, ETTs assist in ventilation of the lungs and may prevent aspiration, umbilical catheters may be used for administration of fluids or medications and for blood sampling, and NGTs may be used for nutritional support, aspiration of gastric contents, or decompression of the gastrointestinal tract in critically ill neonates [3]. Because catheters and tubes (all referred as catheters in the following for simplicity) are typically placed without real-time image guidance, they are frequently malpositioned [15, 12], and serious complications can arise as a result [3]. Thus, the position of a catheter is usually assessed using X-ray imaging immediately following placement [3].

Paediatric radiologists are trained to accurately accomplish the task of detecting catheters on X-ray images and assessing placement with a low diagnostic error rate [10]. However, availability of expertise may be limited or delayed due to high image volumes. An automatic approach is desired to flag X-rays which may have a malpositioned catheter so that they can be immediately reviewed

1st Conference on Medical Imaging with Deep Learning (MIDL 2018), Amsterdam, The Netherlands.

by a clinician or radiologist, thus promoting safer use of catheters. Since the location of a catheter impacts clinical decision making, we believe detection of catheters is a critical first step towards a fully automatic catheter placement evaluation system.

Automated catheter detection is a challenging task. Although most catheters have a radiopaque strip to facilitate detection, the strip may become less apparent depending on the projection angle. Catheters maybe confused by other similar linear structures like ECG leads and anatomy including ribs. Additionally, portions of catheters can be occluded by anatomical structures given that radiographs are a 2D projection of a 3D structure. For example, when a NGT is placed within the oesophagus, the catheter itself becomes less apparent due to the high density of the adjacent vertebrae. Finally, the number and type of catheters that could possibly appear in pediatric X-rays are unknown a priori. The catheters may be intertwined with each other thus making simple line tracing methods fail. Figure 1 gives three sample pediatric X-ray images with some common catheters highlighted in different colors.

Previous methods have heavily relied on primitive low level cues and made superficial assumptions of catheter appearance and position. These works were typically applied to only one or two catheter types and patient positions with limited generality. Machine learning, especially deep learning, has recently received significant attention in the medical imaging community due to its demonstrated potential to complement image interpretation and augment image representation and classification. For example, super human performance has been achieved in organ segmentation in adult chest X-rays [5] and an algorithm is able to denoise low dose computed tomography with improved overall sharpness [27]. All the advances achieved so far have used accumulated annotation datasets. However, in segmentation tasks, the desired pixel level accurate annotated maps are not always available. This is partly because the annotation task requires a certain amount of medical expertise and manual marking is inherently tedious, particularly for objects with elongated structures.

To alleviate this annotation problem in catheter detection, we proposed to use X-ray images with simulated catheters by exploiting the fact that catheters are essentially tubular objects with various cross sectional profiles. To be more specific, a synthetic 2D projection of a catheter is generated by first simulating a horizontal catheter profile and then using it as a brush tip to draw along a B-spline path. This generated catheter is then composited with an X-ray image serving as the training data. Another contribution of this work is a segmentation network that can inherently take into account multi-scale information. This network adopts a UNet-style form and contains a recurrent module that can process inputs with increasing scales[1]. We have empirically shown that by iterating through the scale space of the input image, higher recall is achieved as compared to using a single scale. Details about the methods are discussed in Section 3. Three sample detection results are shown in Figure 1.

## 2   Related Works

There has been limited prior publications regarding automated catheter detection on X-ray images. In this section we not only review catheter detection methods but also provide a brief overview of elongated structure detection as a broader concept.

**Catheter detection** Kao et al. [13] proposed a system to detect ETT on pediatric chest X-rays. It was based on the presumption that ETT usually has highest intensity and is continuous in the cervical region. This system is sensitive to the positioning of the neonates and it is possible to confuse an ETT with a NGT when the assumption no longer holds. Sheng et al. [20] proposed a method for identification of ETT, NGT and feeding tube together in adult chest X-rays. The detection was based on the Hough transform assuming that tubes in small rectangular areas are straight. Their algorithm would fail if the catheter forms a loop. Keller et al. [14] proposed a semi-automated system to detect catheters in chest X-rays with users supplying initial points for catheter tracking. Line tracing was accomplished by template matching of catheter profiles. Mercan et al. [19] proposed a patch based neural network to detect chest tubes and a curve fitting approach to connect discontinued detected line segments. A very recent work used a fully convolutional neural network for detection of peripherally inserted central catheter (PICC) tip position on adult chest X-ray images [16]. A similar approach was taken by Ambrosini et al. [2] to detect catheter in X-ray fluoroscopy but using a UNet-style [18] network. Both methods require human to manually annotate catheter locations for supervised training.

---

[1]Our code is available at https://github.com/xinario/catheter_detection.git.

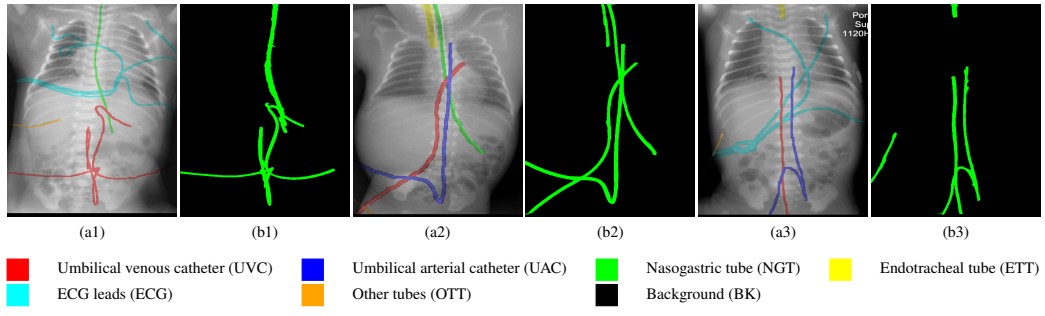

| | | | |
|---|---|---|---|
| 🟥 Umbilical venous catheter (UVC) | 🟦 Umbilical arterial catheter (UAC) | 🟩 Nasogastric tube (NGT) | 🟨 Endotracheal tube (ETT) |
| 🟦 ECG leads (ECG) | 🟧 Other tubes (OTT) | ⬛ Background (BK) | |

Figure 1: Detection of catheters is challenging on pediatric X-ray images. The number of catheters is not known prior to interpretation and they can be partially occluded by the body. ECG electrode leads and other unidentified catheters also serve as sources of confusion. (a1), (a2) and (a3) show the original pediatric X-rays, with the potential catheters, wires and lines (including ECG wires and other unidentified catheters) highlighted in different colors. (b1), (b2) and (b3) show the detected catheters by our proposed method.

**Elongated structure detection** One of the most common elongated structures in medical imaging is a blood vessel. Its detection has been researched in many imaging modalities, such as in retinal fundus imaging [17] and angiography [9]. The methods used in the literature have evolved from hard coded rule based methods into machine learning based methods. In the early days, researchers tried to devise metrics to measure the "vesselness" directly from feature sources like Hessian matrix [9] and co-occurrence matrix [24]. Later on, rather than relying on a single feature, researchers started to aggregate features from multiple sources, such as ridge based [22], wavelets [21] and then employed a supervised learning method on top to delineate the decision boundary between the vessel and non-vessel. Most recent progress was achieved by supervised deep learning where features were directly learnt from images without the intervention of domain expertise [17]. Since blood vessels are of various diameters by nature, multi-scale approaches have also been explored in the literature [28].

## 3 Methodology

### 3.1 Datasets

The training dataset comes from the Open-i dataset [7] from National Institutes of Health (NIH) which contains 7,471 adult chest X-rays. We randomly selected 2515 frontal view images and generated synthetic catheters on them.

The test dataset is collected locally and only contains frontal chest-abdominal X-rays from patients < 4 weeks old. This is the most common radiograph obtained to confirm placement of catheters such as UACs and UVCs in neonates. Currently, the test set has 35 fully labeled images with different catheter types with sample images previously shown in Figure 1. All the annotated catheters (lines excluding ECG leads) are treated as the same class in the detection.

### 3.2 Preprocessing

The X-ray images are of various contrast due to different acquisition protocols. Rather than making the network learn a contrast invariant feature, we normalized the contrast of the input X-rays before sending them for training by using contrast limited adaptive histogram equalization [29] as was done in other works.

### 3.3 Synthetic catheter generation

Catheters are essentially tubular objects, a portion of which is made of radiopaque material with a higher attenuation component designed for ease of detection. Figure 2 (a) shows a simplified cross section profile. This profile would work for both NGT and ETT, as the only difference lies in the catheter width. Using a parallel beam geometry, the projected sinogram is obtained and shown in

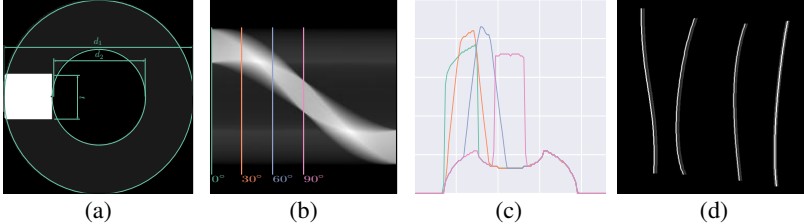

Figure 2: Simulation of catheters in 2D. (a) Simulated cross section profile. (b) Projection profile from $0°$ to $180°$. (c): Projection profile sampled at $0°, 30°, 60°, 90°$. (d) Simulated catheter trace in 2D with the corresponding profile in (c).

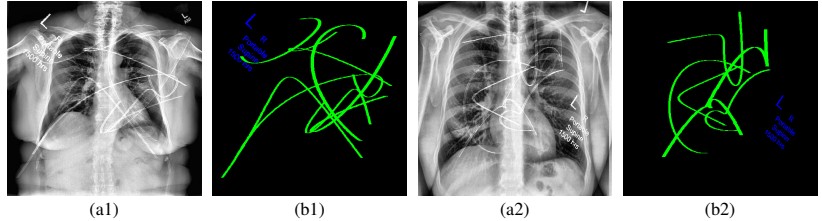

Figure 3: Exemplar training image pairs for the proposed catheter detection network. (a1) and (a2) An adult chest X-ray with synthetic catheters overlaid on the image. (b1) and (b2) The annotation mask used for supervised training.

Figure 2 (b). (c) and (d) are the sampled projection profile at $0°, 30°, 60°, 90°$ and the synthetic catheters drawn with the corresponding profile. Note that the profile used to draw has to be resampled to accommodate the input image size. There are five parameters that are used to parameterize the simulated catheter, inner and outer catheter width, $d_1$ and $d_2$, attenuation coefficients of the catheter and radioopaque material $c_1$ and $c_2$, and the thickness of the strip $t$. A similar approach is used for umbilical venous and arterial catheter (UVC and UAC) but with a profile of dual edges. The tracing of the catheter was simulated using a B-spline with control points randomly generated on the image. De Boor's algorithm [6] was employed for the generation and the generated line was then rasterized with Xiaolin Wu's antialiasing line drawing algorithm [25]. Implementation details can be found in Section 4.1.

### 3.4    Text mask generation

Our initial experiments showed that training with just synthetic lines would cause confusion for radiopaque markers (letters) which may occasionally be noted on radiographs and also share line like structures. Therefore, we explicitly created another class for text so that its misclassification can be penalized independently. For the sake of simplicity, we cropped the common text from the pediatric X-rays and randomly scaled and merged with the adult chest X-ray.

The generated catheter and text are then added to the adult chest X-ray with a weight sampled in the range of 0.15 to 0.35. Figure 3 shows two samples from our synthetic dataset.

### 3.5    Network architecture

Given an input image, the network has to learn to assign each pixel to one of three classes $\{c_{bg}, c_{catheter}, c_{text}\}$. A scale recurrent neural network [23] was employed for this task. It is comprised of an encoder-decoder architecture with shuttle connections and recurrent modules. The encoder progressively increases the number of feature channels and decreases the spatial size (height, width) of the feature map to achieve a certain degree of translation invariance and save memory. The decoder in turn performs an inverse operation to gradually recover the size of the input. During the encoding and decoding process, every single pixel in the final output feature map contains information computed from a large portion of the image hence encodes the global information. The shuttle connection directly communicates lower level features to the higher level so that the network

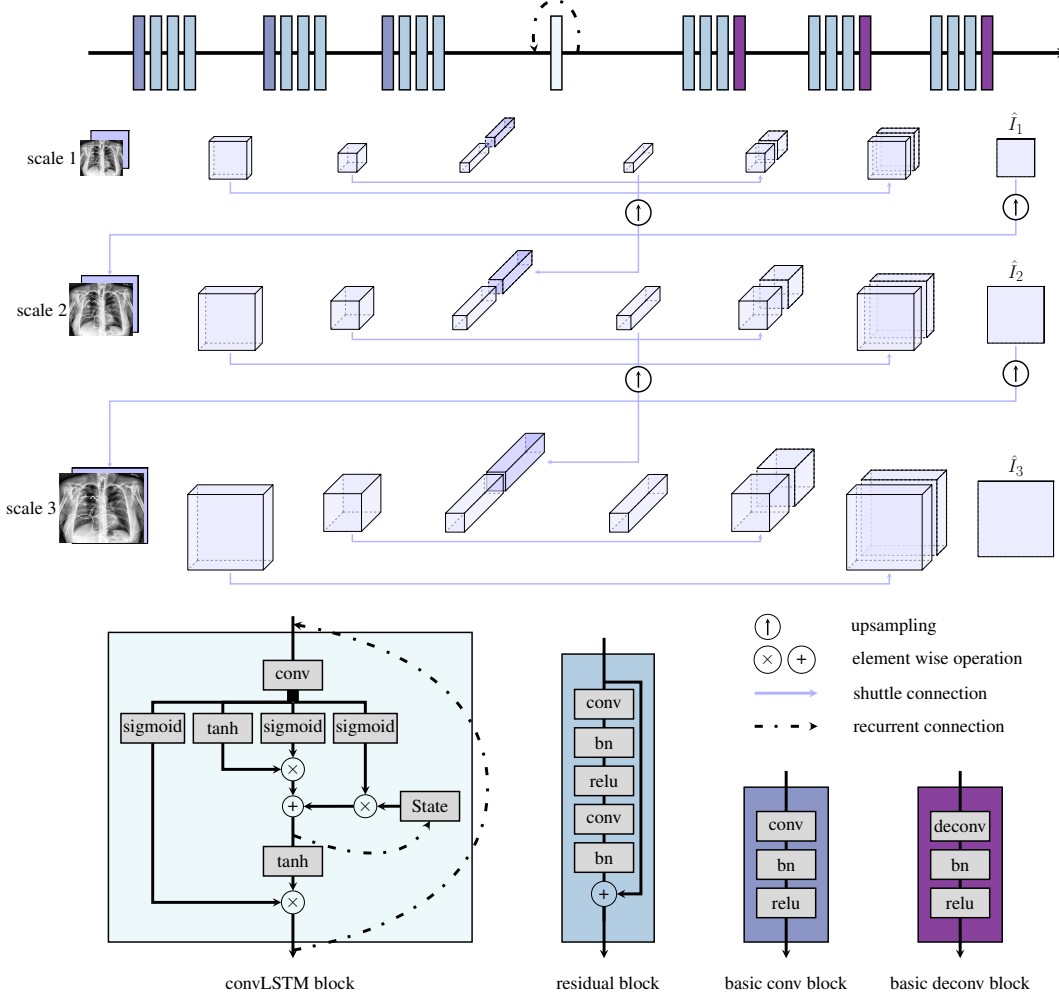

Figure 4: Overview of the network architecture. Note that for the last deconv block, bn and relu were replaced with a softmax layer to get a multi-channel likelihood map.

can make final predictions based on a fusion of both local and global cues. The network is fully convolutional thus can accept images of different scales. Input of increasing scale was sent to the network at different time steps. The recurrent module takes the form of convolutional long-short term memory (convLSTM) [26]. It takes concatenated inputs from the current and previous scale. To maintain size compatibility, we upscaled the feature maps from the previous scale with strided convolution.

Figure 4 provides a general overview of this architecture. Residual block was used to facilitate the training process. Both the skip connection and the residual block [11] benefits training by making the gradient propagate more easily through the network.

## 3.6 Training Objective

The output of the network is a multi-channel feature map with the number of channels equal to the number of predicted classes. We normalized the feature maps with a softmax function so that each channel of the map can be interpreted as the likelihood of belonging to each class. Cross entropy (CE) loss was used to measure the difference between the output and the groundtruth. Loss at each scale was aggregated together as the final optimization objective, which can be expressed mathematically as:

$$\mathcal{L} = \sum_{i=1}^{m} CE(\hat{I}_i, I_i, ; w), \tag{1}$$

where $\hat{I}_i$ is the output of the network at scale $i$ and $I_i$ is the corresponding groundtruth label map. $m$ is the number of the scale and was chosen as 3 in this work. $w$ is the weights to balance the unequal distribution of $\{c_{bg}, c_{catheter}, c_{text}\}$ and was chosen as 1, 40, 80 respectively.

## 4 Experiment setup

### 4.1 Implementation details

The images from the Open-i dataset all have a width of 512. This size was found to be sufficiently large to discriminate different catheters. For NGT and ETT, $d_1$ and $d_2$ were selected as 160 and 80 while $c_1$ and $c_2$ were set as 0.1 and 1, and $t$ was set to be 30 pixels. Note that in the current implementation, the size of $d_2$ was not varied to cope with the width difference of NGT and ETT. For UAC and UVC, only one projection profile at $0°$ was selected. The generated catheter width is 9, 9 and 6 pixels for NGT, ETT, UAC and UVC respectively to accommodate image size. During the training time, the training image pairs were augmented with rotation (in the range of [-60°, 60°]), horizontal flipping, and scale changes (in the range of [0.5, 1.1]) to generate random training image on the fly. Due to the scale change, the augmented images were cropped or padded to the size of $512 \times 512$. The testing images collected locally were all resized to a width of 480 and denoised using BM3D [4] with $\sigma = 0.1$.

The segmentation network was trained on the Cedar cluster of Compute Canada with a P100 GPU. Adam optimizer [35] with $\beta_1 = 0.9$ and $\beta_2 = 0.999$ was used for the optimization with a initial learning rate of 0.0001. The learning rate decayed by 0.1 every 10 epochs. The network was trained to convergence after 50 epochs. All training parameters were initialized with numbers drawn from a Gaussian distribution $\mathcal{N}(0, 0.02)$. Batch size was chosen to be 2 due to the constraint of the GPU memory size.

### 4.2 Evaluation Metrics

In the evaluation, pixels of background and text were treated as the negative class and pixels of catheter were treated as the positive class. Since the class is highly imbalanced, precision and recall were computed with each expressed mathematically as:

$$\text{Precision} = \frac{\text{TP}}{\text{TP} + \text{FP}}, \ \text{Recall} = \frac{\text{TP}}{\text{TP} + \text{FN}} \tag{2}$$

where TP, TN, FP, FN represents the number of true positives, true negatives, false positives, false negatives respectively. The threshold for computing the precision and recall curve was sampled within the range of 0 to 255 at an interval of 30.

Another measure we used for the evaluation is the weighted harmonic mean of precision and recall (or $F_\beta$-measure) which is defined as:

$$F_\beta = \frac{(1 + \beta^2) \times \text{Precision} \times \text{Recall}}{\beta^2 \times \text{Precision} + \text{Recall}} \tag{3}$$

where $\beta^2$ is a weighting term and was set as 0.3 to weight precision more than recall as in [1]. The threshold in calculating the corresponding precision and recall was an image dependent value defined as:

$$T_{seg} = \frac{2}{W \times H} \sum_{x=1}^{W} \sum_{y=1}^{H} \hat{I}^k(x, y) \tag{4}$$

where, $W, H$ are the width and height of the obtained catheter likelihood map $\hat{I}^k$ (assuming at the $k$-th channel of the network output).

### 4.3 Experiments

No prior method is applicable to detect all the catheters of interest, therefore we only compared our method with another deep learning approach which used fcn8s [18] for PICC line tip detection [16]. Further, in order to demonstrate the effectiveness of the recurrent module, we trained another network termed w/oR with the recurrent module removed under the exact same settings. This network resembles the typical UNet-style network used in [2] .

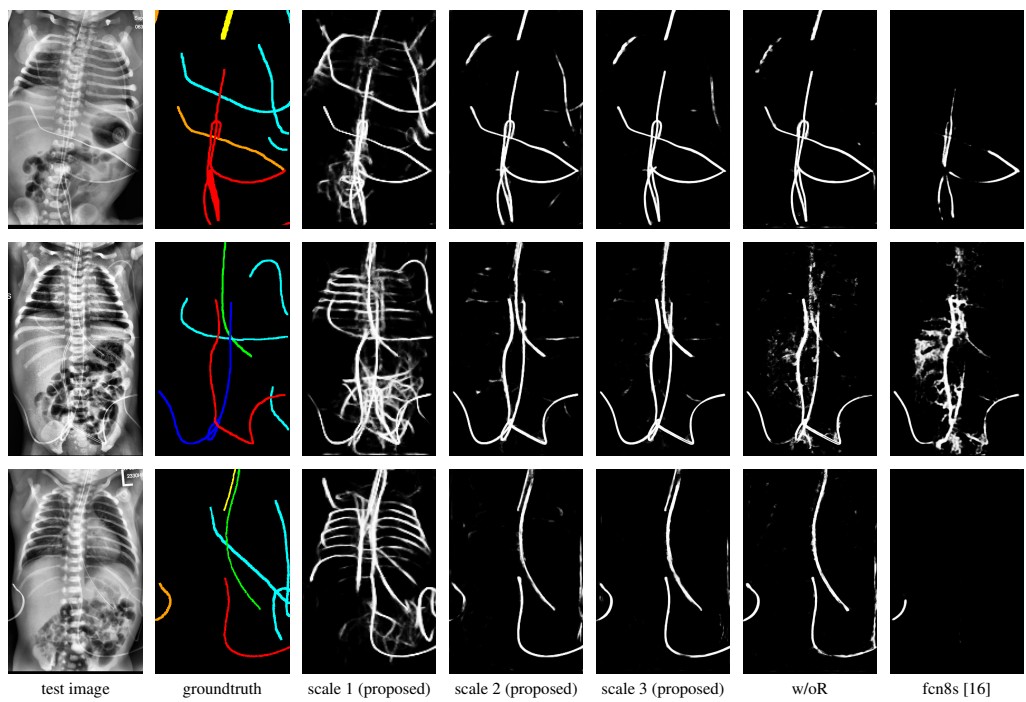

| test image | groundtruth | scale 1 (proposed) | scale 2 (proposed) | scale 3 (proposed) | w/oR | fcn8s [16] |

Figure 5: Raw catheter likelihood maps for different networks on test images: proposed, w/oR, and fcn8s (best viewed in digital version).

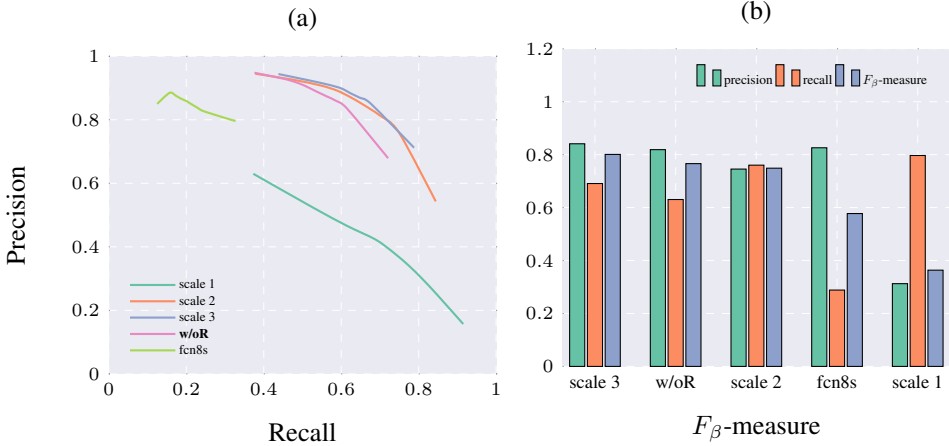

Figure 6: Quantitative results for different methods on the pediatric X-ray test set. (a) Precision and recall curves. (b) $F_\beta$-measures (methods ordered according to the value of the $F_\beta$-measure).

## 5   Results and Discussion

Qualitative visual examples of the raw catheter likelihood maps obtained directly from the network without any postprocessing are shown in Figure 5. It can be seen that the proposed network at the highest scale (scale 3) achieves the best visual appearance as compared to the other methods. The maps from the proposed network at scale 2 and scale 3 look much cleaner than w/oR and fcn8s. We would attribute this to the iterative refinement of the detection results by using the recurrent module. When comparing results from the proposed network at different scales, we can see that the likelihood map from the smallest scale contains almost all line-like structures, including not only catheters but also ribs and ECG leads. This is because catheters, ribs, ECG leads look similar at a smaller scale. These irrelevant line-like structures are gradually filtered out in higher scales because

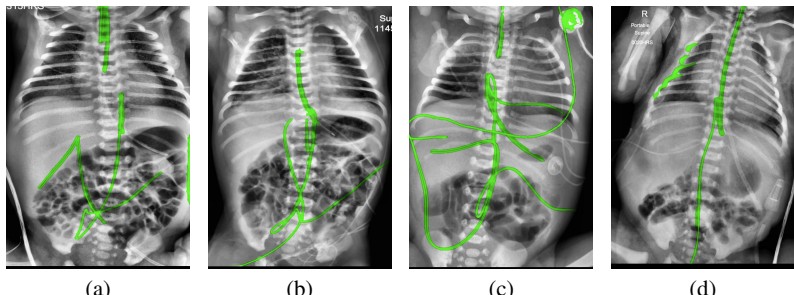

|  (a)  |  (b)  |  (c)  |  (d)  |

Figure 7: Typical failure cases. (a) and (b) partially detected NGT possibly resulted by its similarity to ECG leads. Occlusion of UVC. (c) Confusion with other lines on the X-ray image. (d) Confusion with vertical the lateral aspect of the rib cage (best viewed in digital version).

catheters, especially UVCs and UACs, begin to appear as two parallel edges whereas ribs and ECG leads continue to appear as a single solid line.

Precision and recall curves are shown in Figure 6 (a) and $F_\beta$-measures computed from adaptive threshold are shown in Figure 6 (b). Note that before computing these quantitative measures, the obtained binary map underwent morphological operations to filter out small irregular regions. It can be clearly seen that the proposed method achieves the highest precision and recall than the competitors. The results at lowest scale have the highest recall but lowest precision. The results at the two higher scales achieve approximately the same performance. The reason we believe is that even though there are some improvements in the raw likelihood map, the middle scale has already detected all majority parts of the catheter. The local refinement is too small to be manifested in the quantitative measures. Nonetheless, the $F_\beta$-measure for proposed method at the highest scale ranks the first among the comparators.

Catheters are represented as thin lines of just a few pixels wide on X-ray images. Therefore, a slight pixel shift in the groundtruth annotation could adversely impact the quantitative results. This could inevitably happen due to the nature of this annotation task and we believe our method could provide assistance for annotators in the future by detecting line candidates in the first place.

There are certain situations where our proposed method would fail. Figure 7 (a) and (b) show a partially detected NGT. This mostly likely resulted from the decreased visibility of the radiopaque strip. Figure 7 (a) also shows another failure situation where the inferior portion of the UVC is occluded by the abdomen. (c) shows the case of a falsely detected unidentified line and (d) shows part of the lateral aspect of the rib cage falsely identified as a catheter.

## 6   Conclusion

In this work, we have proposed a simple catheter synthetic approach and a scale recurrent network for catheter and tube detection. Catheters were simulated by using a horizontal projection profile drawn over a randomly generated B-spline. The proposed network could refine the segmentation results by iterating through the scale space of the radiograph input. We have shown that just by training on adult chest X-rays with synthetic catheters, the detection network achieved promising results on real pediatric chest/abdomen X-rays. Although we have experimented with only pediatric X-rays, we believe the methodology should be also applicable to adult X-rays provided the profile is carefully designed with consideration given to the large variation of catheter and wire types. The approach described in this work may contribute to the development of a system to detect and assess the placement of catheters and tubes on X-ray images, thus providing a solution to triage and prioritize X-ray images which have potentially malpositioned catheters for a radiologists urgent review, and ensuring patient safety by alerting the clinician in a timely manner.

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
