# OpenReview forum: "Automatic catheter detection in pediatric X-ray images using a scale-recurrent network and synthetic data"
_MIDL.amsterdam/2018/Conference — MIDL 2018 Poster_

### Review · AnonReviewer3 · 2018-05-04
**simple pipeline, significant amount of experiments**

**Rating:** 3
**Confidence:** 2

**Review:**

This paper introduces synthetic data augmentation and a fully convolutional architecture augmented with a convolutional RNN that operates a different scales to address the problem of catheter detection in pediatric X-ray images. Results are compared on recent methods for catheter detection and baselines.

pros
+ paper well structured, easy to follow
+ simple pipeline, significant amount of experiments

cons
- no comparison with real annotated datasets

* The paper is well written and easy to follow. The motivation of the paper is to exploit synthetic data (synthetic catheters on real X-rays) as a way to augment data.
* Stacking convolutional LSTMs on top of fully convolutional networks has already been introduced for other applications; it might be worth reviewing the model literature as well.
* It seems that synthesized catheters are added to adult X-rays; however the trained model is then applied to segment catheters on patients < 4 weeks old. I am wondering whether this could someone influence the results, could the authors comment on that?
* By shuttle connections, do the authors mean (long) skip connections?
* It would be interesting to perform an analysis on how incrementing the number of scales (from 1 to N) impacts the performance of the system.
* Why not concatenate all previous scales in the ConvLSTM?
* How are parameters d1, d2, c1 and c2 set? What is their influence?
* Figure 3 (b) y-axis can be cut at 1.
* What's the effect of the post-processing step? Do some methods/scales benefit more than other from it?
* It would be worth comparing the performance of using synthetic data vs. real data. If labeled data is available, augmenting the data with synthetic available may still help.



**Special Issue:**

No

---

### Review · AnonReviewer2 · 2018-05-09
**This paper presents the catheter detection method in x-ray images using scale recurrent network. In order to collect training data, authors have constructed synthetic ground-truth dataset. The problem is well-defined and problem-solving approaches are interesting. However, scale-recurrent network which is key idea in this paper didn’t show a significant performance improvement.**

**Rating:** 4
**Confidence:** 2

**Review:**


Quality & Clarity

#1. This paper is well organized, and there is a good problem definition.
#2. In order to solve the lack of training data, authors were described the method clearly.
#3. The description of dataset and experiment design are well written.

Originality & Significance

(+) Training dataset generation method is major novelty of this paper.
(-) But, I’m not sure that the generation method is a suitable approach. Although various types of data have been generated, there is a limit in generating actual data by simply overlaying the synthetic data on an x-ray image.
(+) In order to combine local and global cues and handle various scale, authors have proposed scale-recurrent network.
(-) However, the performance improvement based on the proposed recurrence is insignificant.


**Special Issue:**

No

---

### Review · AnonReviewer1 · 2018-05-10
**Catheter detection trained on synthetic data achieves promising results but not clearly presented**

**Rating:** 3
**Confidence:** 2

**Review:**

The authors propose a multiscale recurrent neural network for segmenting catheters in pedatric X-ray images (adopted from [26]). The method is trained on a large synthetic dataset and applied to real-world data.  The model is evaluated against [16] and a U-net network similar to [2].

I enjoyed reading this paper and the achieved accuracy seems promising. However, I have some concerns with the clarity of the presentation.

* The convLSTM block should be mathematically defined. It is hard to infer the details from Figure 4.

* Figure 4 merges three different graphs but the relationship between the first and second one are not clear to me. There are crossing lines in the graph which obfuscate the dependencies between the boxes (see vicinity of upsampling operator). The predictions are referred to as "I_hat" even though "I" is typically be used for images, i.e., inputs and not predictions.

* How were the hyperparameters (c_bg, c_catheter, c_text, d_1, d_2, c_1, c_2) selected?

* How was the training performed when using a batch size of 2? Was the gradient averaged over multiple iterations? Were there any stability problems with the presented approach?

* How sensitive is the method to hyperparameter settings (e.g., c_bg, c_catheter, c_text)?

* What are the confidence intervals for the given results?



**Special Issue:**

No

---

### Decision · Program_Chairs · 2018-05-15
**Paper7 Acceptance Decision**

Poster